# Responses of Aerial and Belowground Parts of Different Potato (*Solanum tuberosum* L.) Cultivars to Heat Stress

**DOI:** 10.3390/plants12040818

**Published:** 2023-02-12

**Authors:** Jinhua Zhou, Kaifeng Li, Youhan Li, Maoxing Li, Huachun Guo

**Affiliations:** 1College of Agronomy and Biotechnology, Yunnan Agricultural University, Kunming 650201, China; 2Root and Tuber Crop Research Institute, Yunnan Agricultural University, Kunming 650201, China

**Keywords:** potato, heat stress, different plant parts, tuber development

## Abstract

The mechanism of potato (*Solanum tuberosum* L.) thermotolerance has been the focus of intensive research for many years because plant growth and tuber yield are highly sensitive to heat stress. However, the linkage between the aerial and belowground parts of potato plants in response to high temperatures is not clear. To disentangle this issue, the aerial and belowground parts of the heat-resistant cultivar Dian187 (D187) and the heat-sensitive cultivar Qingshu 9 (Qs9) were independently exposed to high-temperature (30 °C) conditions using a special incubator. The results indicated that when the belowground plant parts were maintained at a normal temperature, the growth of the aerial plant parts was maintained even when independently exposed to heat stress. In contrast, the treatment that independently exposed the belowground plant parts to heat stress promoted premature senescence in the plant’s leaves, even when the aerial plant parts were maintained at a normal temperature. When the aerial part of the plant was independently treated with heat stress, tuberization belowground was not delayed, and tuberization suppression was not as severe as when the belowground plant parts independently underwent heat stress. Heat stress on the belowground plant parts alone had virtually no damaging effects on the leaf photosynthetic system but caused distinct tuber deformation, secondary growth, and the loss of tuber skin colour. Transcriptome analysis revealed that the treatment of the belowground plant parts at 30 °C induced 3361 differentially expressed genes in the Qs9 cultivar’s expanding tubers, while the D187 cultivar had only 10,148 differentially expressed genes. Conversely, when only the aerial plant parts were treated at 30 °C, there were just 807 DEGs (differentially expressed genes) in the D187 cultivar’s expanding tubers compared with 6563 DEGs in the Qs9 cultivar, indicating that the two cultivars with different heat sensitivities have distinct regulatory mechanisms of tuberization when exposed to heat stress. The information provided in this study may be useful for further exploring the genes associated with high-temperature resistance in potato cultivars.

## 1. Introduction

The potato (*Solanum tuberosum* L.) is the food and vegetable crop with the highest production in the world, and the consumable part of the potato is called the tuber. Tuberization is affected by many environmental factors, such as light, temperature, and moisture [1,2,3]. In recent years, with the intensification of the greenhouse effect, heat stress has become one of the main factors that endanger potato growth. Studies have shown that if climate change is not alleviated, potato yield is expected to decrease by 30% by 2050 [4]. In addition to yield loss, heat stress also affects the morphological characteristics, physiological and biochemical processes, and the transcriptional regulation of potatoes to varying degrees [5].

*SELF-PRUNING 6A* (*StSP6A*), a gene that controls potato tuber formation, is an orthologue *FLOWERING LOCUS T* (*FT*) in *Arabidopsis thaliana*. The elevated expression of *StSP6A* in potato leaves and tubers is a precursor of tuber formation. The overexpression of *StSP6A* leads to early tuber formation, while the inhibition of *StSP6A* expression by RNAi completely prevents tuber formation [6]. The expression of *StSP6A* is negatively regulated by *SELF-PRUNING 5G* (*StSP5G*), another member of the *FT* family; *StSP5G* is activated by its upstream genes *CONSTANS LIKE 1* (*StCOL1*) (a homologue of *CONSTANS*) and *PHYTOCHROME B* (*StPHYB*) [7], and the expression of *StCOL1* is inhibited by *CYCLING DOF FACTOR* (*StCDF1*) [8]. In addition to the regulatory pathway, the overexpression of *GA2 oxidase 1* (*StGA2ox1*) reduces gibberellin content and thus promotes potato tuber formation, while the overexpression of *GA20 oxidase 1* (*StGA20ox1*) inhibits potato tuber formation [9,10]. Tuber formation is also regulated by *BELLRINGER-1 LIKE 5* (*StBEL5*) and *POTATO HOMEOBOX-1* (*StPOTH1*), two transcription factors, whose mRNAs are transported from leaves to stolons via the phloem [11]. They are also involved in hormone metabolism and the transcriptional activation of *StSP6A* and *StCDF1*, which promotes potato tuber formation [12,13]. In addition, microRNAs have been shown to control tuber formation. For example, *micRNA172* promotes tuber formation by upregulating the expression of *StBEL5* [14], and *micRNA156* inhibits tuber formation by inhibiting the expression of *micRNA172* and *StSP6A* [15].

In recent years, studies have found two important factors that regulate tuber development under high-temperature conditions. The two factors inhibit tuber formation by inhibiting *StSP6A* expression. Morris et al. (2019) experimentally found that the *TIMING of CAB EXPRESSION 1* (*StTOC1*), an upstream regulator of *StSP6A*, specifically bound to the promoter region of *StSP6A* and that heat increased the expression of *StTOC1*, thereby inhibiting the expression of *StSP6A*. They also found that silencing *StTOC1* restored the expression of *StSP6A* at high temperatures to normal levels so that potato plants grew tubers normally [16]. Lehretz et al. (2019) found that high temperature caused a small RNA that was *suppressing the expression of SP6A* (*SES*) to accumulate and inhibit *StSP6A* expression and that after *SES* was inhibited by the short tandem target mimic (STTM) method, *StSP6A* expression and tuber formation remained normal at a high temperature [17]. Recently, Park et al. (2022) reported that the level of *StSP6A* transcription is repressed by various regulatory pathways in the early and late stages of heat stress, with posttranscriptional regulation in the early stage and transcriptional repression in the later stage [18].

The organs or tissues of whole plants can sense increasing heat. However, plant aerial parts are more directly exposed to high temperatures. Unlike rice, corn, and tomatoes, whose edible parts are harvested aboveground, the harvested potato organs are located belowground. Therefore, it is necessary to study the mechanism of potato thermotolerance or heat response in both the plant’s aerial and belowground parts. To date, most previous studies on heat stress in potato have focused on the whole plant. It is unknown whether high temperature inhibits potato tuber formation via the aerial part of the plant, the belowground part of the plant, or the mutual interaction between the aerial and the belowground parts of the plant. In this work, we performed four treatments on potatoes, namely a high-temperature treatment on the aerial part alone (AH), a high-temperature treatment on the underground part alone (UH), a high-temperature treatment on the entire plant (EH), and a normal-temperature treatment on the entire plant (EN), to clarify the plant part(s) through which high temperature inhibits potato tuber formation and to determine the underlying mechanism. The results provide a theoretical basis for understanding how potato plants will cope with heat stress in the future.

## 2. Materials and Methods

### 2.1. Potato Genotypes and Growth Conditions

The test potato materials were provided by the Root and Tuber Crop Research Institute of Yunnan Agricultural University, including the heat-sensitive cultivar Qingshu 9 (Qs9) and the heat-tolerant cultivar Dian 187 (D187).

The terminal buds of the potato tissue culture seedlings with consistent growth were transferred to MS medium for 20 days for cultivation (16 h light/8 h dark, light intensity 100 μmol·m^−2^·s^−1^, temperature 20 ± 1 °C). Thereafter, the tissue culture seedlings were transplanted into heat-sterilized matrix soil and placed in an artificial climate box for 20 days (16 h light/8 h dark, light intensity 100 μmol·m^−2^·s^−1^, temperature 20 ± 0.5 °C, relative humidity 75%). The plants with consistent growth were selected for the experimental treatments and cultivated in an artificial climate box at 12 h light/12 h dark, a light intensity of 200 μmol·m^−2^·s^−1^, a relative humidity of 45%, and a temperature of 20 ± 0.5 °C until the stolon formation stage. Next, plants with similar growth potential were divided into 4 groups (40 plants in each group), and the plants in the various treatment groups were subjected to different temperature treatments (Figure 1). (1) In EN (20/20 °C), the entire plant was cultivated at normal temperature (20 ± 0.5 °C). (2) In UH (20/30 °C), the belowground part of the plant was cultivated at a high temperature by placing the nutrient pots in a thermostatic water pool at 30 ± 0.5 °C, and the aerial part of the plant was cultivated in an incubator at 20 ± 0.5 °C. (3) In EH (30/30 °C), the entire plant was cultivated at a high temperature (30 ± 0.5 °C). (4) In AH (30/20 °C), the aerial part of the plant was cultivated in an incubator at a high temperature (30 ± 0.5 °C), and the belowground part of the plant was cultivated at 20 ± 0.5 °C by placing the nutrient pots in a thermostatic water pool at 20 ± 0.5 °C. Except for the different temperatures, the environmental factors were constant among the 4 treatment groups (illumination duration, light intensity, and humidity were 12 h, 200 μmol·m^−2^·s^−1^, and 45%, respectively). The plants were watered every day during the treatment period to avoid drought stress and ensure that the relative soil moisture content was 80% of the field capacity.

To carry out the experiment smoothly, we modified the artificial climate box. Specifically, a small pool was placed in the artificial climate box, and the water in a temperature-controlled water tank and a small pool was continuously circulated through a water pump, ensuring that the water temperature in the small pool was uniform with the temperature-controlled water tank. Next, the nutrient pot was placed in the small pool so that the soil temperature of the nutrient pot reached that of the water temperature of the pool, thereby changing the ambient temperature of the belowground part of the potato plants (Figure 1).

### 2.2. Measurements of the Morphological Index, Stolons, and Tubers

Here, 10 plants were selected from each treatment group for morphological index measurements before the start of the treatment, on day 7 of the treatment, on day 14 of the treatment, and at the end of the treatment (on day 49 or 77 of the treatment). The plant height was measured from the base of the plant to the growth point at the top with a ruler. The basic leaf angle (i.e., the angle between the stem and the petiole of the fourth lowest leaf) and the leaf drooping angle (i.e., the angle between the main stem and the line connecting the apex and the petiole base of the fourth lowest leaf) were measured with a goniometer. The internodal length, which refers to the distance between two adjacent nodes, was measured with a ruler, and the mean internodal length was calculated. The number of leaves per plant, which refers to the number of green and healthy compound leaves of a potato plant, was counted. Finally, the number of nodes per plant was counted.

After the morphological index measurements from each sampling period were completed, the number of stolons, the number of tubers, and the tuber yield of each plant were determined, and the tubers were classified by weight into 11 grades (<1 g, 1–2 g, 2–3 g, 3–4 g, 4–5 g, 5–6 g, 6–7 g, 7–8 g, 8–9 g, 9–10 g, >10 g).

### 2.3. Measurements of Photosynthetic Gas Exchange

On day 14 of the treatment (the EN-, UH-, and AH-treated plants all entered the tuber-formation stage), the parietal leaflets of the fourth lowest leaf were selected to measure the leaf-gas exchange parameters by using an Li-6400XT photosynthesis and fluorescence system (Li-Cor, Lincoln, NE, USA). The net photosynthetic rate (P_n_) and the transpiration rate (T_r_) were measured directly by the instrument, and the dark respiration rate (R_d_) and the photorespiration rate (R_L_) were calculated on the basis of the measured light response curve and CO_2_ response curve according to the method of Bassman et al. (1991) [19].

### 2.4. Total RNA Isolation and Sequencing

On day 14 of the treatment, the fourth lowest leaf and the newly formed tubers (or stolons) of the plants were collected, nine plants were randomly selected from each treatment group, and three plants were mixed as a biological replicate. The sampling time was noon on day 14 of the treatment (i.e., after 6 h of illumination). The samples were quickly frozen in liquid nitrogen, 100 mg portions were taken for total RNA extraction by the TRIzol method, and the purity and the concentration of RNA were assessed with a Nanodrop ND-2000 instrument (Thermo Scientific, Allentown, PA, USA). Qualified RNA samples were delivered to Shanghai Applied Protein Technology Co., Ltd. (Shanghai, China) for cDNA library construction and sequencing on the NovaSeq6000 sequencing platform.

### 2.5. Wayne Analysis, Heatmap Analysis, and GO and KEGG Enrichment Analysis

The filtered clean reads of the original sequencing data were compared with the reference genome (DM1-3_v6.1) by using HISAT2 software (http://daehwankimlab.github.io/hisat2/ (accessed on 7 December 2022)) to obtain mapped reads. Next, the fragments per kilobase of transcript per million mapped fragments (FPKM) value of each gene was calculated by using Feature Counts (http://subread.sourceforge.net/ (accessed on 7 December 2022)), and the FPKM value was used to make a heatmap by using TBtools software to analyse the expression pattern of the genes related to tuber formation. Finally, the differentially expressed genes (DEGs) between the treatment groups, which were genes satisfying |log2FC| > 1 and Padj < 0.05, were screened by using DESeq2 (http://bioconductor.org/packages/release/bioc/html/DESeq2.html (accessed on 7 December 2022)).

The obtained DEGs were subjected to Wayne analysis with TBtools software. We hypothesized that because AH- and UH-treated plants formed tubers, they would have a similar expression pattern as the genes regulating tuber formation in the EN-treated plants but a different expression pattern from that of EH-treated plants. Therefore, we compared AH- and UH-treated plants with EN-treated plants, extracted the DEGs, and then carried out a Wayne analysis to obtain the common DEGs between the two groups, namely the common DEGs between the tuber-forming treatments. Similarly, we compared AH- and UH-treated plants with EH-treated plants, extracted the DEGs, and then performed a Wayne analysis to obtain the common DEGs between the two groups, namely the common DEGs between the tuber-forming treatments and the nontuber-forming treatment. Finally, the common DEGs between the tuber-forming treatments and the common DEGs between the tuber-forming and nontuber-forming treatments were subjected to a Wayne analysis again, and after excluding the common results between the two, the remaining common DEGs between the tuber-forming treatments and the remaining common DEGs between the tuber-forming and nontuber-forming treatments were taken to be the specific DEGs not affecting tuber formation and the specific DEGs affecting tuber formation, respectively.

GO and KEGG enrichment analyses of the specific DEGs obtained by Wayne analysis were performed using the GO and KEGG analysis plugins, and the pathway enrichment map was generated in R.

### 2.6. Quantitative Real-Time PCR Analysis

A gene expression analysis was performed on an ABI 7500 RT-qPCR instrument by the SYBR Green I chimeric fluorescence method according to the manufacturer’s instructions. The primers used in this study are shown in Appendix A, where *Elongation Factor 1-alpha* (*StEF1α*) was applied as a reference gene. The RNA-Seq raw data in our study were submitted.

### 2.7. Data Processing

Excel 2019 was used for data processing, and graphs were plotted with GraphPad Prism 8. The statistical analysis included an analysis of variance (ANOVA) using the SPSS software package (Chicago, IL, USA). Before hypothesis testing and relationship analysis, a data normality test was performed. ANOVA results were considered significant at *p* < 0.05, and the mean comparisons were performed by using Duncan’s multiple range test. Additionally, the least-square means, standard deviation, variance, and descriptive statistics, such as the coefficient of variation, range, skewness, and kurtosis, were estimated. The data were expressed as the mean ± standard deviation. Correlation coefficients between the RT-qPCR expression and FPKM values were calculated by the univariate linear regression method at a significance level of *p* < 0.05, and the determination index R^2^ was determined.

## 3. Results

### 3.1. The Plants under Heat Stress Showed Enhanced Vertical Growth but Weakened Lateral Growth

The heights of the EH- and AH-treated plants significantly increased during the duration of the experiment and were always significantly higher than those of the EN- and UH-treated plants (Figure 2a,b). In the first and second weeks of treatment, there was no significant difference between the heights of the EN- and UH-treated plants or between the heights of the EH- and AH-treated plants; at the end of the treatment, the EH-treated plants had the greatest height, which was significantly taller than that of the other three treatments, and the AH-treated plants were significantly taller than the EN- and UH-treated plants (no significant height difference was observed between the EN- and UH-treated plants) (Figure 3a,d). To determine the reasons for the significant differences in plant height, we also measured the number of nodes and the internodal length. The results indicated that the findings for internodal length were consistent with those for plant height throughout the treatment period (Figure 3c,f). In contrast, the number of nodes was significantly different only between treatment groups at the end of the treatment, but there was a slight difference between the treatment groups in the first 2 weeks of treatment (Figure 3b,e). The results show that the short-term heat stress increased the potato plant height mainly thanks to the growth of internodal length rather than an increase in the node numbers and that the long-term heat stress increased the potato plant height thanks to the joint results of the internodal length growth and the increase in node numbers. The UH treatment did not have a significant effect on the potato plant height, while the AH treatment increased the effect of the heat stress on the potato plant height under the long-term treatment compared with the EH treatment. These findings indicate that maintaining a suitable temperature for the belowground parts of the plant can reduce the excessive growth of the aerial plant parts caused by high temperature and that the response to heat stress in the aerial plant parts is regulated by the belowground plant parts.

In contrast to the characteristics of plant height, the basic leaf angle and leaf drooping angle of the EH- and AH-treated plants were always significantly lower than those of the EN- and UH-treated plants and significantly declined during the duration of the treatment period (Figure 2a,b). The basic leaf angle and leaf drooping angle were not significantly different between the EN- and UH-treated Qs9 plants (Figure 3h,i) or between the EH- and AH-treated D187 plants (Figure 3k,l). In addition, in the first 2 weeks of treatment, the number of healthy leaves of the EH-treated Qs9 plants was significantly lower than that under the other three treatments, while the number of healthy leaves of the D187 plants was not significantly different among the treatments. By the end of the treatment, the EH treatment resulted in the highest numbers of healthy leaves in the Qs9 and D187 plants, which were significantly higher than those in the other three treatments (Figure 3g,j), and the leaves of the EN- and UH-treated plants were senescent and turned yellow. Notably, the degree of the leaf senescence of the UH-treated plants was significantly higher than that of the EN-treated plants (Figure 2a,b). This finding indicates that the UH treatment promoted premature senescence in the potato plants, while the high-temperature treatments prolonged the growth period, and the promoting effect was most significant in the EH treatment group. The differences in plant height further demonstrate that the changes in the aerial plant parts are regulated by the belowground parts.

### 3.2. Delayed Potato Tuber Formation Due to Heat Stress

The numbers of stolons in the EN- and AH-treated Qs9 plants were significantly lower than those of the EH-treated Qs9 plants after 1 week of treatment, and the number of stolons of the UH-treated plants was significantly lower than that of the EH-treated plants after 2 weeks of treatment. By the end of the treatment, only the EH-treated plants still had stolons, and almost all the stolons in the remaining treatments had turned into tubers (Figure 4a,c). In contrast to the stolons, tubers were formed in the EN- and AH-treated Qs9 plants after 1 week of treatment, tuber formation started in the UH-treated Qs9 plants after 2 weeks of treatment, and tuber formation in the EH-treated Qs9 plants had just started at the end of the treatment (Figure 4a,d,e).

The number of stolons in the D187 plants decreased over time, and at the observation time points, there was no significant difference among the four treatments. By the end of the treatment, only the EH-treated plants still had stolons, and the stolons of the plants under the other treatments were almost all transformed into tubers (Figure 4b,f). Similar to the Qs9 plants, the EN- and AH-treated D187 plants had formed tubers after 1 week of treatment, the UH-treated plants showed only weak tuber formation after 2 weeks of treatment, and the EH-treated plants did not start tuber formation until the end of the treatment (Figure 4b,g,h). Notably, some stolons in the UH-treated Qs9 and D187 plants transformed into shoots (Figure 4a,b). The above analysis demonstrates that high temperature delayed the formation of potato tubers, and the delay effect of the treatment groups followed the descending order of EH > UH > AH.

### 3.3. Significant Reduction in Potato Yield under Heat Stress

By the end of the treatment (on day 49 of the treatment), the EN-, UH-, and AH-treated Qs9 plants formed obvious tubers, while the EH-treated Qs9 plants formed only smaller tubers. In addition, the UH treatment caused tuber deformation, secondary growth, and the loss of skin colour (Figure 5a). Among the four treatments, the EN treatment resulted in the highest yield per plant, which was significantly higher than that under the UH and AH treatments (the AH treatment resulted in significantly higher yields than the UH treatment did); the EH treatment resulted in the lowest yield (Figure 5c). The UH treatment resulted in the highest number of tubers per plant, but the number of tubers per plant in the UH treatment group was not significantly different from that of the EN or AH treatment groups (Figure 5d). To understand the effect of heat stress on different parts of potato plants by the weight distribution of potato tubers, we classified the harvested tubers into 11 grades according to weight and calculated the proportion of each grade to create a heatmap. The results demonstrated that the EN treatment resulted in the widest distribution of tuber weights, followed by the AH and UH treatments, and the EH treatment resulted in the narrowest distribution of tuber weights; the EH-treated plants formed the smallest tubers, all less than 1 g (Figure 5e).

At the end of the four treatments (on day 77 of the treatment), the D187 plants all formed obvious tubers (Figure 5b). The results verified that D187 is a heat-tolerant cultivar and that Qs9 is a heat-sensitive cultivar. However, the UH treatment still led to tuber deformation and secondary growth in the D187 plants. There were significant differences in the yield per plant among the treatment groups, and the yield per plant followed the descending order of EN > AH > UH > EH (Figure 5c). The EH treatment resulted in the fewest tubers per plant, which was significantly lower than the numbers of tuber under the other three treatments, but there was no significant difference among the other three treatments (Figure 5d). The tuber weight distribution range of the D187 plants was similar to that of the Qs9 plants (Figure 5e).

Hence, heat stress significantly reduced the yield of the potato plants and even completely prevented tuber formation, depending on the treatment group. However, maintaining either the aerial plant parts or the belowground plant parts at a suitable temperature while the other part was treated with heat stress allowed the potato plants to form tubers, which indicates that soil temperature has a greater impact than air temperature on tuber formation.

### 3.4. There Was No Significant Difference in Photosynthetic Parameters between the Aerial Plant Parts at the Same Temperature

As the production organ of assimilation products, leaves are directly related to tuber development. Therefore, we measured the photosynthetic gas exchange parameters to further analyse the physiological state of leaves under different high-temperature treatments. Table 1 shows that in the second week of treatment, the P_n_ of the EH-treated Qs9 plants was the lowest and was significantly lower than that of the other three treatments, and there was no significant difference among the other three treatments. In contrast, the P_n_ of the D187 plants was not significantly different among the four treatments. The aerial parts treated at the same temperature showed no significant difference in R_d_, R_L_, T_r_, or water-use efficiency (WUE), whereas the aerial parts treated at high temperature had higher R_d_, R_L_, and T_r_ but lower WUE than the aerial parts treated at normal temperature. The results indicate that high temperatures increase the respiration consumption of potato plants, which may be an important reason for the delayed tuber formation and the significant yield drop.

### 3.5. Transcriptional Regulation of Potato Tuber Formation under Heat Stress

The previous analysis demonstrates that after 2 weeks of treatment, the EN-, AH-, and UH-treated plants formed tubers, but the EH-treated plants did not form smaller tubers until the end of the treatment. We suspected that a large number of genes must be involved in causing the large differences in tuber development. Therefore, we collected potato leaves and tubers (stolon tips from the EH-treated plants) after 2 weeks of treatment for transcriptome sequencing.

The sequencing results indicated that the overall alignment rate of all the biological replicates with the reference genome exceeded 83.03% (Appendix A), which indicated that the sequencing quality met the requirements for the subsequent analysis. Principal component analysis divided the leaves and tubers (or stolons) of the different treatment groups into four independent groups (Appendix A), which indicated that the high-temperature treatments of the different plant parts caused different gene expression patterns among the four treatments.

According to the Wayne analysis method, the Qs9 plants had 188 specific DEGs not affecting tuber formation and 306 specific DEGs affecting tuber formation in the leaves of the Qs9 plants and had 1123 specific DEGs not affecting tuber formation and 2880 specific DEGs affecting tuber formation in the tubers (or stolons) (Figure 6a). The D187 plants had 110 specific DEGs not affecting tuber formation and 339 specific DEGs affecting tuber formation in the leaves and had 421 specific DEGs not affecting tuber formation and 658 specific DEGs affecting tuber formation in the tubers (or stolons) (Figure 6b). It is worth noting that there were more specific DEGs affecting tuber formation than specific DEGs not affecting tuber formation in the leaves or tubers (or stolons), which confirmed our previous conjecture that a large number of genes are involved in the heat inhibition of tuber formation.

Next, we performed GO and KEGG enrichment analyses on the specific DEGs affecting tuber formation in the leaves and tubers. The GO enrichment analysis revealed that the specific DEGs affecting tuber formation in the tubers of the Qs9 plants are significantly enriched in biological processes such as starch and carbohydrate metabolism and temperature response (Appendix A) and that the specific DEGs affecting tuber formation in the leaves and tubers of the D187 plants are significantly enriched in biological processes such as carbohydrate metabolism and temperature (high temperature) response (Appendix A). In addition, these DEGs are also significantly enriched in some of the pathways of cellular components and molecular functions, such as enzyme activity, cell walls, and amyloplasts. The KEGG enrichment analysis showed that the specific DEGs affecting tuber formation in the leaves and tubers of the QS9 and D187 plants are significantly enriched in the starch and sucrose metabolic pathways (Appendix A), while the specific DEGs not affecting tuber formation are not significantly enriched in these pathways (Appendix A). Tuber formation is closely related to carbohydrate and starch metabolism [18]. In this experiment, the KEGG and GO enrichment analyses revealed that the DEGs affecting tuber formation are significantly enriched in these metabolic pathways (Appendix A), which indicates that high temperature hinders the biosynthesis of carbohydrates in leaves and their accumulation in tubers and thus inhibits tuber formation and development.

Previous studies have demonstrated many key genes involved in the regulation of tuber development (Appendix A), and we analysed the expression patterns of these genes. The heatmap indicated that the expression of tuber-formation-promoting genes in both the leaves and tubers (or stolons) of the Qs9 and D187 plants were higher in the tuber-forming treatments than in the EH treatment, whereas the expression patterns of the tuber-formation-inhibiting genes in both the leaves and tubers (or stolons) of the Qs9 and D187 plants were lower in the tuber-forming treatments than in the EH treatment (Figure 7). The RT-qPCR analysis showed that the tuber-formation-promoting genes *StSP6A* and *ADP-GLUCOSE PYROPHOSPHORYLASE* (*StAGPase*) were downregulated in the leaves and stolons of the EH-treated plants relative to the EN-treated plants, and their relative expression was significantly lower than that under the other three treatments. *StGA2ox* was downregulated in the stolons of the EH-treated plants, while *StBEL5* and *TERMINAL FLOWER1* (*StTFL1*) were downregulated in the leaves of the EH-treated plants. *POTATO LIPOXYGENASE* (*StPOTLX*) was downregulated in the stolons of the EH-treated Qs9 plants and in the leaves of the EH-treated D187 plants (Figure 8a,c). Compared with the EN-treated plants, the six key genes inhibiting tuber formation in the stolons of the EH-treated plants were upregulated, and their relative expression was significantly higher than that under the other three treatments (Figure 8b,d). These results indicate that the genes promoting tuber formation at high temperatures can function in both potato leaves and tubers (or stolons), while the genes inhibiting tuber formation function mainly in the tubers (or stolons), showing high tissue specificity.

To verify the transcriptome sequencing results, we selected the RT-qPCR results of eight genes and the sequenced FPKM values for a univariate linear regression analysis. The results indicate that the R^2^ values of all the tested genes exceeded 0.61 (Appendix A), which indicates that the transcriptome sequencing results are correct.

## 4. Discussion

Previous studies have demonstrated that potato-tuber-forming signals are synthesized in the leaf and transported as mobile signals through the phloem to the stolon tip to induce tuber formation [6]. High temperature inhibits the biosynthesis and expression of these tuber-forming signals, thereby inhibiting tuber formation [20]. We therefore hypothesized that tuber formation would not occur under the AH treatment but would occur under the UH treatment. However, the test results showed that tubers were formed under both the AH and UH treatments (Figure 5a,b). These results indicated that the high temperatures inhibited tuber formation in the potato plants if both the aerial and belowground parts underwent heat stress, but as long as either the aerial or belowground parts of the plants were maintained at a suitable temperature while the other part underwent heat stress, the potato plants still formed tubers.

Previous experiments have indicated that compared with the EN treatment, the AH and UH treatments reduced the yield per plant by 96.15% and 92.31%, respectively, and the yield per plant in the AH- treatment group was lower than that of the UH-treated plants [21]. In this study, compared with the EN treatment, the AH and UH treatments reduced the yield per plant of the Qs9 plants by 26.56% and 52.53%, respectively, and the yield per plant of the D187 plants by 38.79% and 63.17%, respectively, which was much smaller than the reduction reported by a previous study; additionally, the yield per plant in the AH treatment group was significantly higher than that of the UH-treated plants (Figure 5c). A possible reason for this difference is that the treatment temperature used by Reynolds et al. (1989) was 34/30 °C (day/night), which is higher than that used in this experiment. In addition, his experimental device employed a spiral copper tube that was inserted into each nutrient pot and relied on the circulation of water kept at a constant temperature to change the soil temperature, which likely caused an uneven distribution of the soil’s temperature near the copper pipes and at the edges of the nutrient pots, thereby affecting the test results.

The UH-treated plants had significantly lower yield per plant and more small tubers and aerial tubers than the AH-treated plants (Reynolds et al. [21] also found this phenomenon in their study), and the UH treatment caused tuber deformation, secondary growth, and the loss of skin colour (Figure 5a,b). These findings demonstrate that although potato plants can form tubers at high temperatures by independently maintaining a suitable temperature for their aerial parts or belowground parts, tuber growth is better maintained if the belowground parts are kept at a suitable temperature. In addition, compared with the Qs9 plants, the D187 plants formed tubers under the EH treatment, and the number of tubers per plant under the EH treatment was only slightly different from those of the other three treatments (Figure 5b,d); this finding indicates that under heat stress, the heat-tolerant potato cultivar has significantly more stable tuber-formation traits and a more-robust tuber-formation ability.

At the end of the heat treatments (on day 49 or 77 of treatment), the AH-, UH-, and EH-treated Qs9 and D187 plants had significantly lower yields than the EN-treated Qs9 and D187 plants, and the EH-treated Qs9 plants had just begun to form small tubers. To further analyse the effects of the different high-temperature treatments on tuber development, we counted the number of stolons and the number of tubers at weeks 0, 1, and 2 of the treatment. Park et al. (2022) used the cultivar Desirée as the test material and exposed the whole plant to high-temperature stress, and they found that a heat treatment of 30/24 °C (day/night) delayed the formation of stolons and tubers by 1 week, while the treatment of 35/29 °C (day/night) completely prevented the formation of stolons and tubers at the 4-week observation period [18]. Similar results were obtained in the present experiment. The AH treatment did not delay tuber formation, the UH treatment delayed tuber formation by 1 week, and the EH treatment delayed tuber formation by 7 weeks (Figure 4a,b). The results suggest that maintaining the aerial parts of potato plants at a suitable temperature can reduce the delayed effect of heat stress on tuber formation and that maintaining the belowground plant parts at a suitable temperature can eliminate the delaying effect of heat stress. These results further indicate that tuber formation depends mainly on the temperature regulation network of the belowground parts of potato plants.

During the experiment, we observed significant changes in the morphology of the potato plants, in addition to the tuber development’s being affected by the differing heat-stress treatments. Therefore, we recorded the plant morphological traits at the same time points that we recorded tuber traits. In the short-term treatments (1 and 2 weeks), there was no significant difference in the morphological characteristics of the aerial parts of the potato plants between the experimental treatments, but in the long-term treatment (7 and 11 weeks), the plant height, number of nodes, internodal length, and number of healthy green leaves of the AH-treated plants were significantly lower than those of the EH-treated plants (Figure 3). This finding indicates that the heat-stress treatment of the belowground parts of the potato plants induced a significant feedback-regulation effect on the morphological characteristics of the aerial plant parts only after an extended period (7 and 11 weeks). In addition, compared with the Qs9 plants, the EH- and AH-treated D187 plants had no significant difference in basic leaf angle or leaf drooping angle during the long-term treatment (7 and 11 weeks) (Figure 3k,l), which indicates that the heat-tolerant potato cultivar can survive long-term heat stress and possesses more stable leaf traits.

Potato cultivars with different heat tolerances show different changes in P_n_ under heat stress, and the P_n_ of heat-sensitive potato cultivars, such as cv. Agria, significantly decrease under heat stress [22], while heat-tolerant potato cultivars, such as Desirée and Norchip, show a slight increase in P_n_ under heat stress [2]. In the present experiment, the EH treatment resulted in significantly lower P_n_ in the Qs9 (heat-sensitive) plants than in the other three treatments, but the P_n_ of the D187 (heat-tolerant) plants was not significantly different among the four treatments (Table 1). The test results of Hastilestari et al. (2018) indicated that the P_n_ of the heat-sensitive potato cultivar Agria under AH, UH, and EH treatments were significantly lower than that under the EN treatment; the T_r_ under the UH treatment was significantly lower than that under the EN treatment; and the belowground parts of the plants showed a significant feedback-regulation effect on the aerial parts [22]. In the present experiment, there was no significant difference between the P_n_ of the EN-treated Qs9 plants and that of the AH- and UH-treated Qs9 plants, and there was no significant difference in R_d_, R_L_, T_r_, or WUE when the aerial plant parts were at the same temperature (Table 1). A possible reason for this finding is that we measured P_n_ at the tuber-formation stage in this experiment, while Rina et al. measured P_n_ at the tuber expansion stage. Different measurement periods resulted in different measurement results. High temperature increases dark respiration in potatoes [23] and photorespiration consumption [24], which reduces the amount of ribulose-1,5-bisphosphate carboxylase (Rubisco) involved in carboxylation, resulting in a weakened dark reaction. The same results were obtained in this experiment. The R_d_ and the R_L_ of the aerial plant parts under the high-temperature treatments were significantly higher than those of the aerial parts under the normal-temperature treatment (Table 1).

One study found that under heat stress, restoring the expression of *StSP6A* in the leaves of potato plants restored the number of tubers per plant to normal levels but could not restore the yield per plant [18], a finding that was confirmed in the present experiment. The expression of *StSP6A* in the leaves of the Qs9 and D187 plants under the UH treatment was higher than that under the other three treatments (Figure 8a,c), and there was no significant difference in the number of tubers per plant between the UH-treated plants and the EN-treated plants (Figure 5d). However, the yield per plant of the UH-treated plants was still significantly lower than that of the EN-treated plants (Figure 5c). Another study found that the overexpression of *StSP6A* led to early tuber formation, a change in the sink–source balance, enhanced sink-absorption capacity, and increased yield [17]. In the present experiment, the expression levels of *StSP6A* in the leaves of the UH-treated Qs9 and D187 plants were higher than those in the EN-treated plants, but the tuber-formation stage of the UH-treated plants was delayed by 1 week compared with that of the EN-treated plants, and the yield per plant of the UH-treated plants was significantly reduced. A possible reason for this difference between the two studies is that the previous study did not measure *StSP6A* expression in tubers (or stolons). In the present experiment, the *StSP6A* expression in the tubers of the UH-treated Qs9 and D187 plants was significantly lower than that of the EN-treated plants; there was no significant difference between *StSP6A* expression in the tubers of the AH- and EN-treated plants (Figure 8a,c), and the tuber-formation time was the same for these two treatments (Figure 4a,b). Hence, *StSP6A* expression affects mainly the tuber-formation stage and the yield.

It has been documented that except for *StTOC1* and *StAGPase*, the abovementioned genes used for the RT-qPCR analysis are involved in the photoperiodic pathway [25]. In the present experiment, the relative expression levels of *StSP6A* and *StAGPase* were consistent with tuberization, while the relative expression levels of the remaining genes were not consistent with tuberization. The expression of tuber-formation-promoting *StTFL1* and *StBEL5* in the stolons of the EH-treated plants was higher than in the three tuber-forming treatments, and the levels of the expression of tuber-formation-inhibiting *StSP5G*, *BELLRINGER-1 LIKE 29* (*StBEL29*), and *SUCROSE TRANSPORTER 4* (*StSUT4*) were not the highest in the leaves of the EH-treated plants (Figure 8), which indicates that the regulatory mechanisms of the photoperiod-dependent tuber-formation pathway are temperature dependent.

*PHYB* acts as the first receptor in the photoperiodic pathway regulating tuber formation and subsequently regulates the expression of other genes, thereby regulating tuber formation. *PHYB* can also act as a temperature sensor. Previous studies have found that the expression of *PHYB* does not significantly change with temperature [22,26]. However, in the present experiment, the expression of *StPHYB* in the leaves and tubers of the Qs9 and D187 plants was significantly different under different temperature treatments, and most notably, the expression of *StPHYB* in the tubers (or stolons) of the EH-treated plants was four times higher than that in the other three treatments (Figure 8b,d). A possible reason for this discrepancy between the findings is that Hastilestari et al. (2018) used a photoperiod of 16 h/8 h (day/night) and a high-temperature treatment duration of 10 days, while we used a photoperiod of 12 h/12 h (day/night) and a high-temperature treatment duration of 14 days.

*StBEL29* and *StGA2ox* have the same function as *StSP6A* and can also directly regulate tuber formation, but *StGA2ox* plays a role mainly in tubers (or stolon tips) and can inhibit the biosynthesis of gibberellins to reduce the gibberellin content in stolon tips, thus inducing tuber formation [9,27]. In the present experiment, the expression of *StGA2ox* in the leaves at high temperature was lower than that at normal temperature, while the expression of *StGA2ox* in the tubers was consistent with tuber formation (the lowest expression in the EH-treated plants) (Figure 8a,c). These results indicate that high temperature affects the expression of *StGA2ox*, but *StGA2ox* functions mainly in tubers (or stolon tips) to regulate tuber formation. In contrast, it has been demonstrated that *StBEL29* can migrate and function in both leaves and tubers (or stolon tips) [28]. However, in the present experiment, *StBEL29* had the highest expression in the stolon tips of the EH-treated plants, which was significantly higher than that in the plants under the other three treatments (Figure 8b,d). These results indicate that although *StBEL29* can migrate, it functions mainly in tubers (or stolon tips) to achieve the temperature-dependent regulation of tuber formation.

*StAGPase* (the starch precursor gene), a key gene in starch biosynthesis, is significantly reduced under drought stress [29]. In this experiment, the expression of *StAGPase* was the lowest in the leaves and tubers (or stolons) of the EH-treated Qs9 and D187 plants, which was significantly lower than that in the other three treatments (Figure 8a,c). This result indicates that *StAGPase* is not only regulated by moisture content but also affected by temperature. It is worth noting that *StAGPase* had the same expression pattern as *StSP6A* in the experiment (Figure 7 and Figure 8a,c) and that KEGG and GO enrichment analyses revealed that the DEGs affecting tuber formation are significantly enriched in the starch biosynthesis and metabolic pathways (Appendix A). Hence, we hypothesize that *StAGPase* may be another key gene regulating tuber formation in a temperature-dependent manner, which can be further studied in the future.

## 5. Summary

Heat stress delays tuber formation in potatoes, thus significantly reducing yield or completely preventing tuber formation. However, maintaining the aerial parts of potato plants at a suitable temperature (while the below parts are independently exposed to heat stress) can reduce the delay of tuber formation caused by heat stress, although the formed tubers will have problems of deformation, secondary growth, and the loss of skin colour. However, maintaining the belowground part at an appropriate temperature (while the aerial parts are independently exposed to heat stress) can eliminate the delay of tuber formation caused by heat stress and weaken the adverse effects of high temperature on potato tubers. High temperature can significantly increase plant height and internodal length and significantly reduce leaf angles. The feedback-regulation effect of the belowground parts of potato plants on the morphology of the aerial part was not significant in the early stage of treatment. The P_n_ of the heat-sensitive potato cultivar significantly decreased under the EH treatment, but the P_n_ of the heat-tolerant cultivar was not significantly different among the four treatments. In addition, there was no significant difference in the T_r_, R_d_, R_L_, or WUE of the aerial parts of the plants at the same temperature. The transcriptional regulation analysis indicated that there were a large number of DEGs between the tuber-forming and nontuber-forming treatments in the tuber-formation period, and these genes are significantly enriched in the biosynthesis and metabolic pathways of sugars, carbohydrates, and starches. It has been confirmed that tuber-formation-promoting genes function in both leaves and tubers (or stolons), while tuber-formation-inhibiting genes function mainly in tubers (or stolons). The results demonstrate that high temperature inhibits tuber formation in potato plants through both the aerial and belowground parts, but maintaining the belowground part at a suitable temperature (while the aerial part is independently exposed to heat stress) can restore the normal expression of key tuber-formation-promoting genes in stolons, thus enabling tuber formation at high temperatures and significantly reducing the adverse effects of heat stress on potato tubers.

## Figures and Tables

**Figure 1 plants-12-00818-f001:**
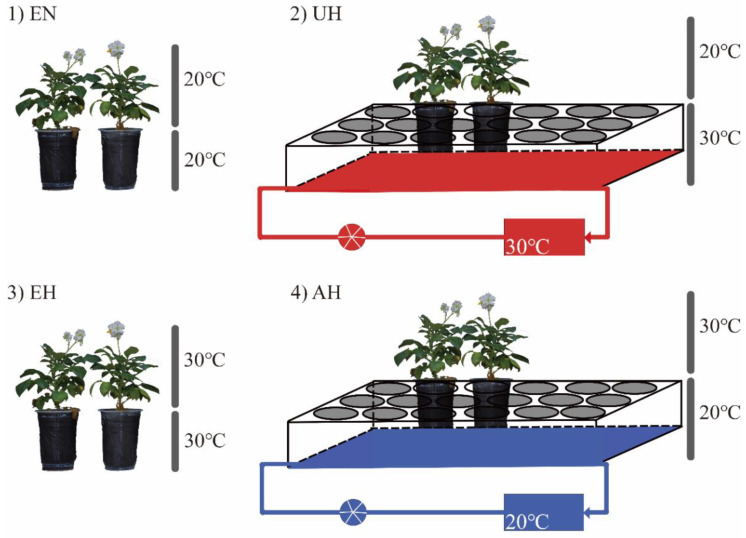
A schematic diagram of the heat-stress treatments for the different potato plant parts. (**1**) EN: a normal-temperature treatment on the entire plant; (**2**) UH: a high-temperature treatment on the underground part alone; (**3**) EH: a high-temperature treatment on the entire plant; (**4**) AH: a high-temperature treatment on the aerial part alone.

**Figure 2 plants-12-00818-f002:**
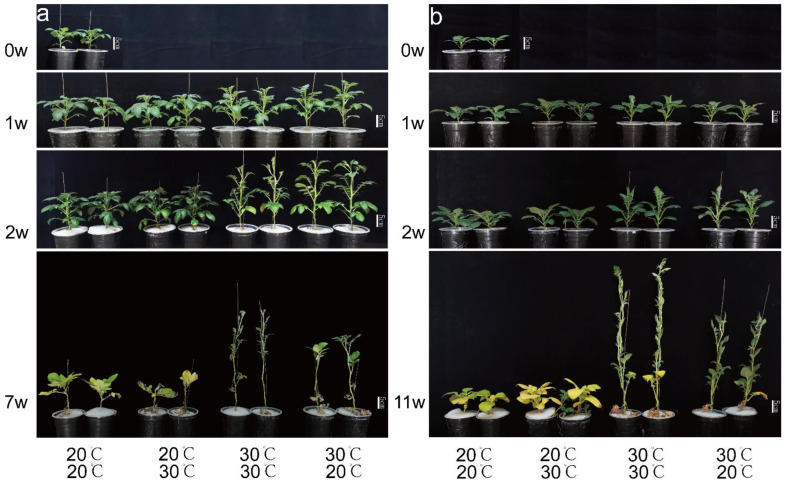
Phenotypes of the aerial plant parts of Qs9 (**a**) and D187 (**b**) under the heat-stress treatment of different plant parts. Potato plants were grown for 0, 1, 2, 7, or 11 weeks at the indicated temperature. Representative pictures among 9 replicates are displayed. w, weeks.

**Figure 3 plants-12-00818-f003:**
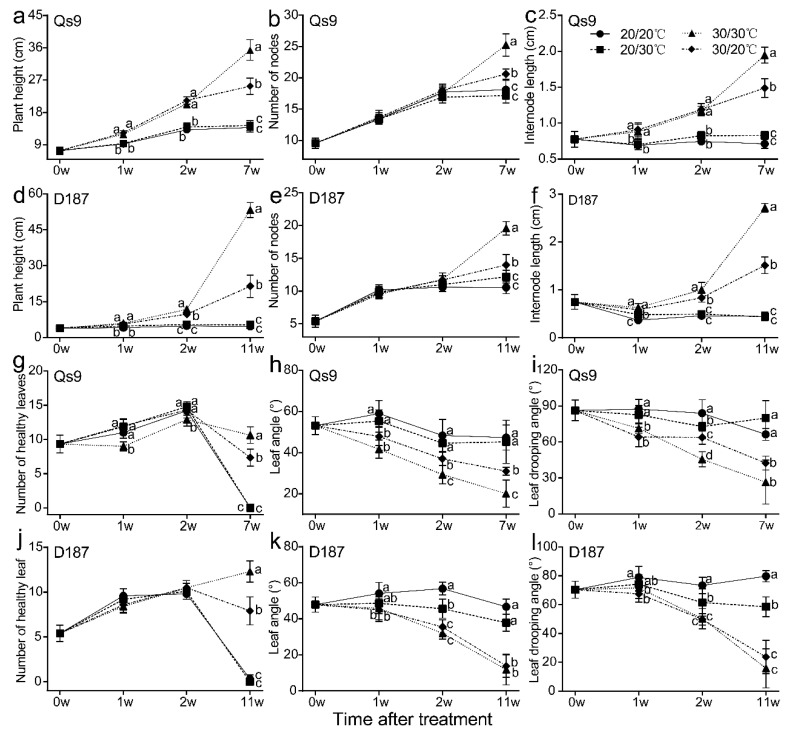
Measurement of stem phenotype and leaf angle under heat-stress treatment of different plant parts. (**a**–**c**) and (**g**–**i**): plant height (**a**), number of nodes (**b**), internode length (**c**), number of healthy leaves (**g**), leaf angle (**h**), and leaf drooping angle (**i**) of the Qs9 plants at the indicated time points during the treatment. (**d**–**f)** and (**j**–**l**): plant height (**d**), number of nodes (**e**), internode length (**f**), number of healthy leaves (**j**), leaf angle (**k**), and leaf drooping angle (**l**) of the D187 plants at the indicated time points during the treatment. In total, 10 replicates were averaged and statistically analysed. The different lowercase letters indicate groups that are significantly different from one another.

**Figure 4 plants-12-00818-f004:**
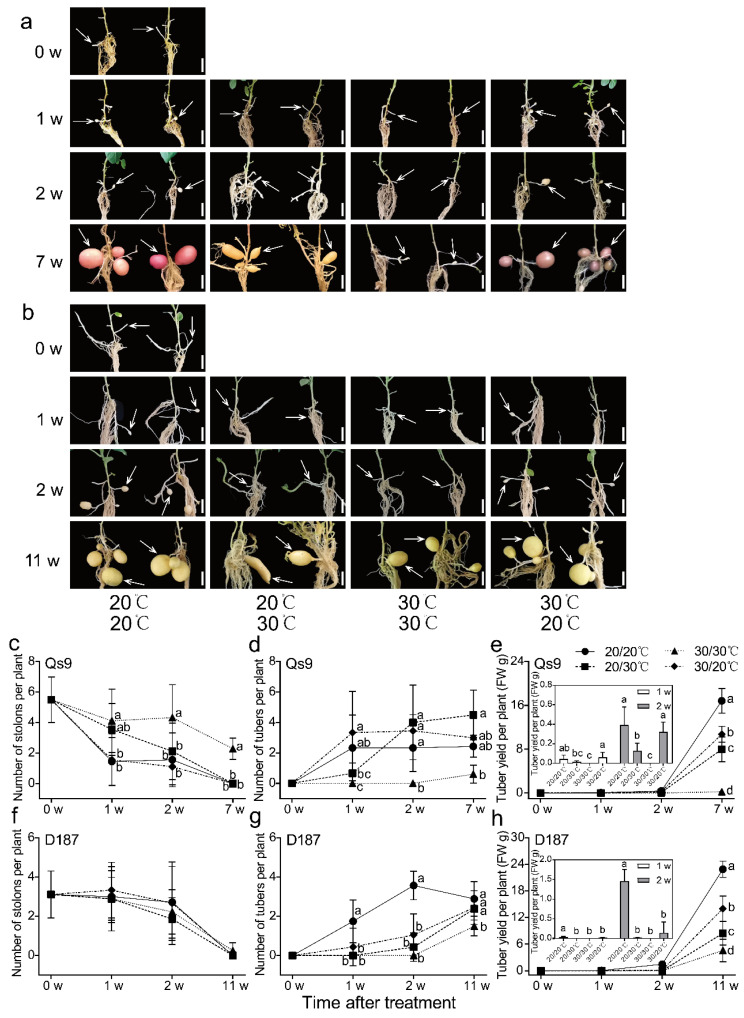
Potato tuberization of the Qs9 and D187 plants under the heat-stress treatment of different plant parts. Potato plants of the Qs9 (**a**) and D187 (**b**) cultivars were grown for 0, 1, 2, 7, or 11 weeks under heat-stress treatments of different parts of the plant. Photographs were taken after the removal of soil. Representative pictures among the nine replicates are displayed. Scale bars, 2 cm. The white arrow points to the tuber or stolon. w, weeks. (**c**–**e**): stolon number (**c**), tuber number (**d**), and tuber yield (**e**) per plant of the Qs9 cultivar at the indicated time points during the treatment. (**f**–**h**): stolon number (**f**), tuber number (**g**), and tuber yield (**h**) per plant of the D187 cultivar at the indicated time points during the treatment. Nine replicates were averaged and statistically analysed. (**e**,**h**) Graph inset showing an enlarged view of the tuber yield at 1 and 2 weeks. The different lowercase letters indicate groups that are significantly different from one another.

**Figure 5 plants-12-00818-f005:**
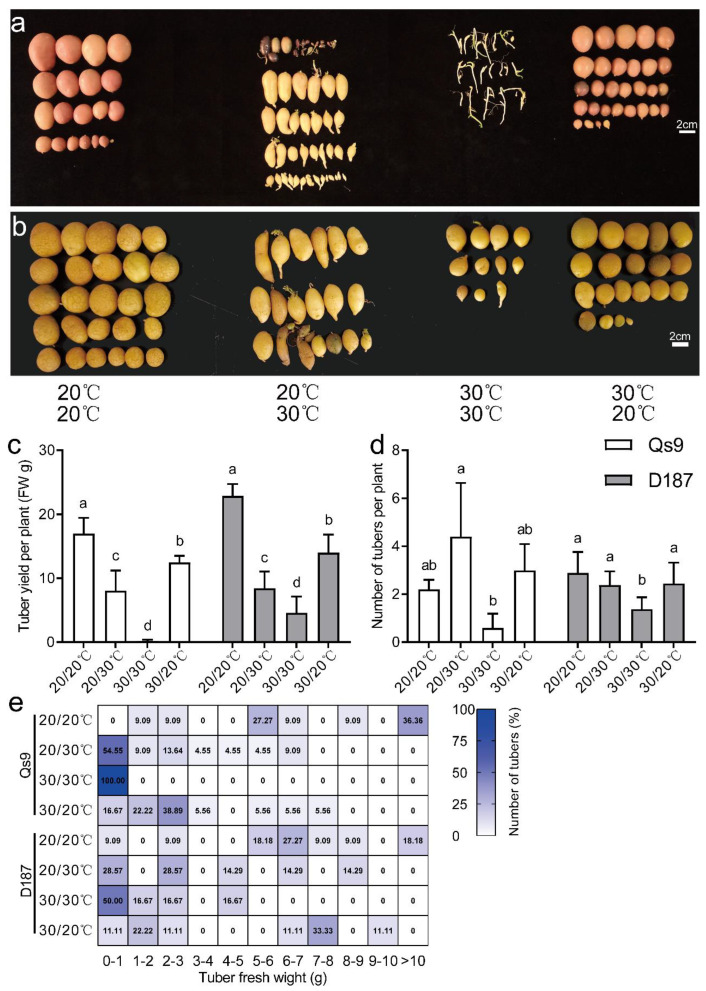
Phenotypes and tuber productivity of the Qs9 and D187 plants under the heat-stress treatment of different plant parts. (**a**,**b**): Photographs of the total tubers harvested from the nine plants. **a**: Qs9, **b**: D187. (**c**,**d**): Measurement of tuber productivity under the heat-stress treatments of different parts of the plants. Tuber yield (**c**) and tuber number (**d**) per plant; nine replicates were averaged and statistically analysed. The different lowercase letters indicate groups that are significantly different from one another. (**e**) The percentage of tuber numbers among the total number of tubers in the annotated range of fresh weight is displayed using a heatmap. The total number of tubers is the sum of the tuber numbers in all analysed potato plants in each treatment.

**Figure 6 plants-12-00818-f006:**
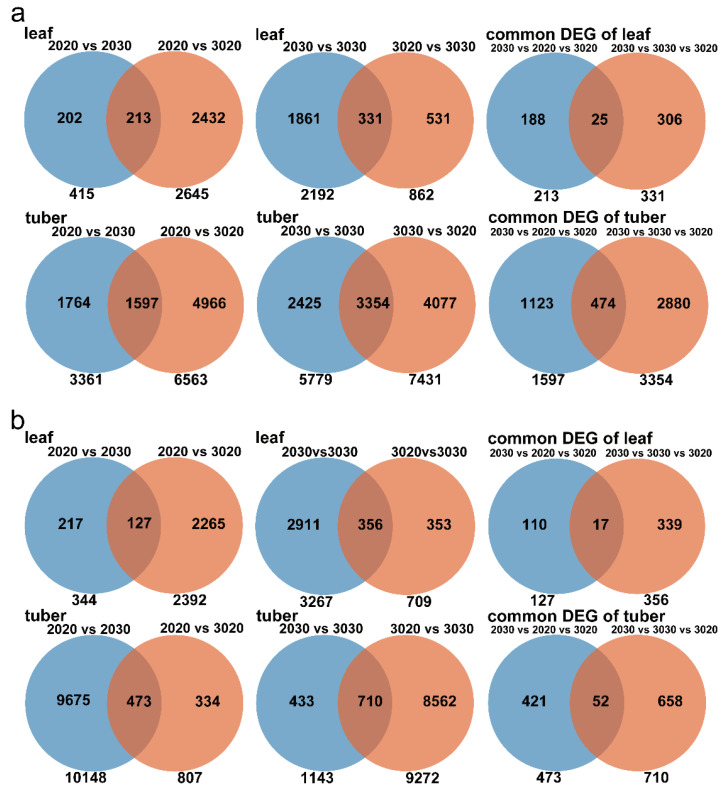
A Venn diagram showing the number of DEGs of the Qs9 (**a**) and D187 (**b**) plants under the heat-stress treatment of different plant parts. Genes with a log2-fold change greater than 1.5-fold and a *p*-value < 0.01 were considered DEGs.

**Figure 7 plants-12-00818-f007:**
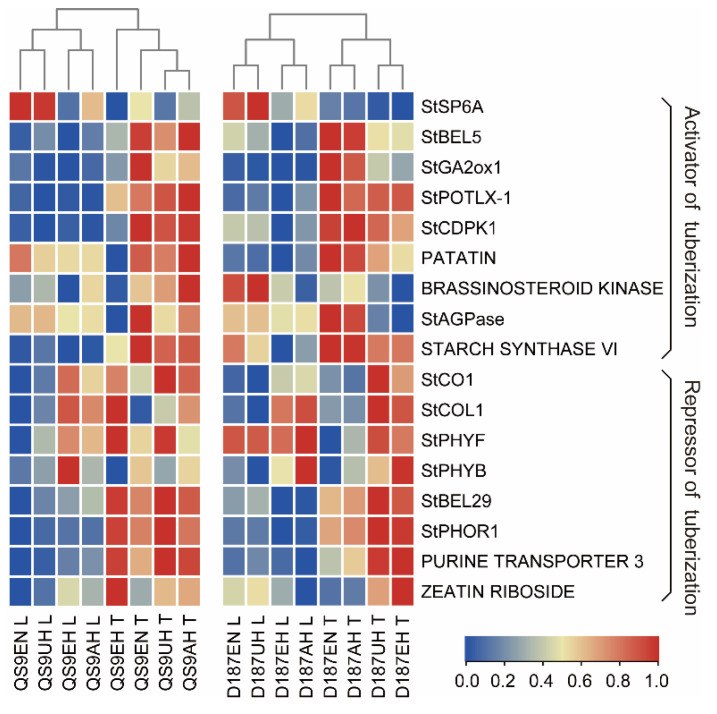
The expression of the selected genes from the transcriptome analysis. Expression of the candidate genes involved in temperature-dependent tuberization. Red to blue indicates expression from high to low. The colour scale represents log_2_ (FPKM+1).

**Figure 8 plants-12-00818-f008:**
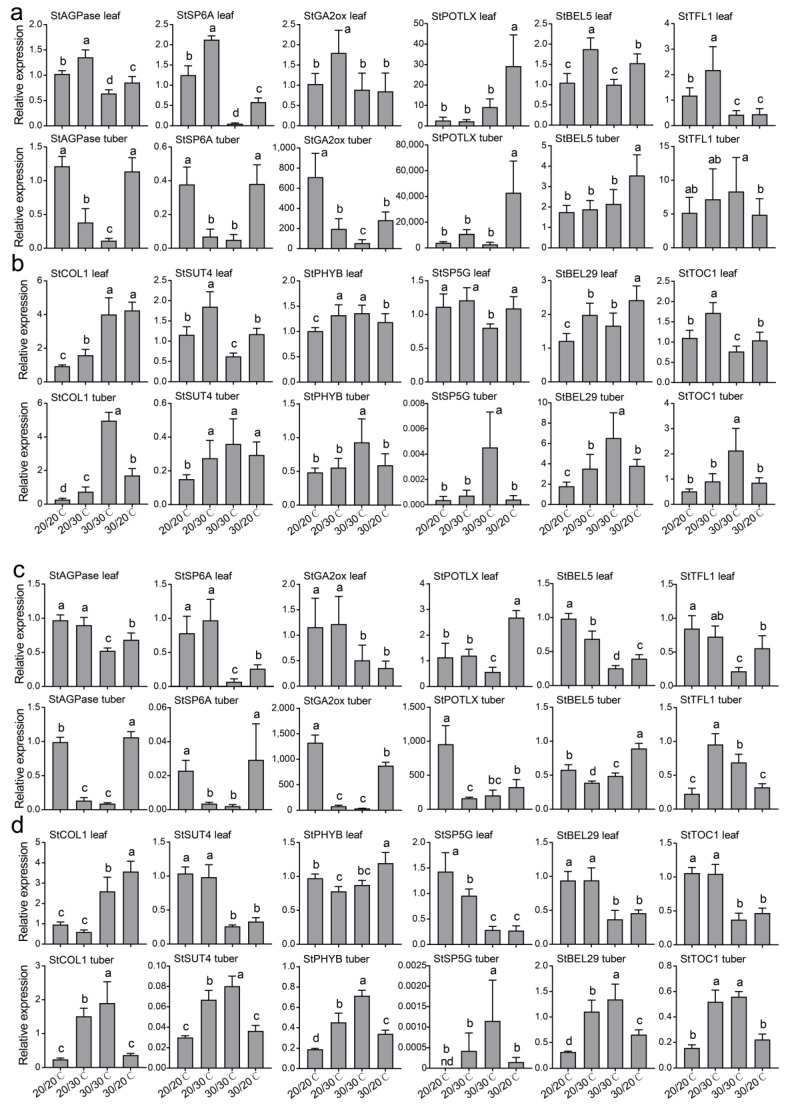
The expression of selected genes from the Qs9 (**a**,**b**) and D187 (**c**,**d**) plants under the heat-stress treatment of different plant parts was determined by RT-qPCR. Activator (**a**,**c**) and repressor (**b**,**d**) genes of tuberization. Three replicates were averaged and statistically analysed. The different lowercase letters indicate groups that are significantly different from one another.

**Table 1 plants-12-00818-t001:** The photosynthetic parameters of the Qs9 and D187 plants under the heat-stress treatment of different plant parts.

Materials	Treatment	Photosynthetic Parameters				
		P_n_ (μmol CO_2_·m^−2^·s^−1^)	R_d_ (μmol·m^−2^·s^−1^)	R_L_ (μmol·m^−2^·s^−1^)	T_r_ (mmol H_2_O·m^−2^·s^−1^)	WUE (μmol·mmol^−1^)
Qs9	20/20 ℃	10.75 ± 0.52 a	−1.18 ± 0.23 a	−2.33 ± 0.11 a	2.48 ± 0.31 b	4.38 ± 0.31 a
	20/30 ℃	11.18 ± 0.62 a	−1.13 ± 0.66 a	−2.82 ± 0.08 a	2.54 ± 0.31 b	4.45 ± 0.48 a
	30/30 ℃	9.83 ± 0.03 b	−2.14 ± 0.37 b	−5.68 ± 0.25 b	8.85 ± 0.22 a	1.11 ± 0.03 b
	30/20 ℃	10.52 ± 0.32 a	−2.33 ± 0.10 b	−4.91 ± 0.44 b	9.55 ± 0.38 a	1.10 ± 0.05 b
D187	20/20 ℃	8.18 ± 0.95 a	−0.82 ± 0.35 a	−2.80 ± 0.09 a	2.21 ± 0.86 b	4.01 ± 0.79 a
	20/30 ℃	7.63 ± 0.98 a	−0.81 ± 0.14 a	−3.05 ± 0.38 a	1.61 ± 0.29 b	4.15 ± 0.25 a
	30/30 ℃	8.37 ± 0.71 a	−1.51 ± 0.61 b	−4.06 ± 0.17 b	6.92 ± 1.24 a	1.24 ± 0.17 b
	30/20 ℃	9.47 ± 0.78 a	−1.73 ± 0.40 b	−4.28 ± 0.18 b	7.48 ± 0.71 a	1.28 ± 0.15 b

Three replicates were averaged and statistically analysed. The different lowercase letters indicate groups that are significantly different from one another.

## Data Availability

All data generated and analysed during this study are included in this article (and the Appendix A).

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
