# Peer review of "Responses of Aerial and Belowground Parts of Different Potato (Solanum tuberosum L.) Cultivars to Heat Stress"

_plants, 2023, doi:10.3390/plants12040818_

Round 1
Reviewer 1 Report
Great job!
Author Response
Thank you very much for your comments
Reviewer 2 Report
The article is quite interesting and deserves attention. However, some points require improvement.
1. The abstract lacks information on cultivars and their physiological and molecular responses to high temperature
2. Please explain why the research was conducted using seedlings. It is known that potato cultivars are highly heterozygous.
3. There is lack information on reference gene for relative expression of the selected genes in RT-qPCR analyses.
4. Description of results needs to be revised. I suggest that the description of the morphological responses of the plants should follow the developmental phases, first plant growth, then tuber formation and tuber productivity. Please include information which genes up- and down-regulated (Figure 9). The effect of high temperature should on relative expression of selected genes should be referred to the control (plants growing at 20°C). I propose that the same gene expression scale be used for each variety.
5. Please explain „The color scale represents log2 changes” (Line 441)
6. The visualization of GO and KEGG analysis of DEGs should be improved.
7. Other comments are included in the text.

Author Response
Please see the attachmen

Reviewer 3 Report
Dear Authors,
Detailed notes on the manuscript are given below:
1) Abstract - in the first sentence, add the purpose of the work,
2) 2.8. Data processin - specify the description of the statistical analysis:
- ANOVA was a parametric version (add that the normality of the distribution of the data population and the homogeneity of variance in the samples were tested - these are the conditions necessary to adjust the parametric ANOVA; what tests, what result (e.g. K-S, Levene ...)),
- add that you used Duncan's test as a post-hoc procedure and based on it you determined groups of homogeneous variables (you can omit this information in the case of charts and tables),
- in Figures 2 and 9 you have shown error bars (write in the methodology what they represent: Sd, average, 5% ...?)
- Figure S3. Linear correlation between ... - add in the methodology that you studied the correlation (what; Pearson? Sperman?), add that you determined the trend line (I suppose the least squares method) and the equations of these lines, add that the determination index R2 was determined,
3) Figure no. 3 - data in the graph (c-h) should not be combined
4) Drawing no. 5 - as above
5) Figure 7. GO and KEGG .... - axis description is illegible
6) "Figure S2. GO and KEGG analysis of DEGs (not affecting tuberization)." - correct the drawing number + note as above
7) Figure S3. Linear correlation between ... note as above
8) 5. Conclusion - I suggest changing it to "Summary"
9) The scientific literature on plant stresses is vast; should be supplemented with significant items. This will also allow you to build the chapter "Introduction" (plants reaction to cold, UV, electromagnetic fields, lack of water, etc ...)
Round 2
Reviewer 2 Report
The description of the results needs further revision. Potato growth and development follows successive developmental phases: plant growth, stolon induction and tuber development. Starting the description of the results with the final phase introduces confusion, the work is not clear and consequently reduces the quality of the presented results. For this reason, I consider it is necessary to change the order in which the results are presented. The Authors also not corrected the description of gene expressions (line 492-505). Please show which genes up- and down-regulated. The influence of high temperature please refer to the control.
The Response 1: „Our study found that Qs9 and D187 had similar changes in morphology, physiology, transcriptional regulation, and expression of key genes of tuber formation in addition to tuberization under high-temperature stress” is surprising considerring the data presented, for example in Figure 2 or Figure 8.
The title „Influence on tuber development in different parts of potato plants under high temperature stress” is badly formulated. Please state the influence of which factor on what proces.
In addition, the quality of figures S2 and S3 is still poor.
Reviewer 3 Report
Thank you for proofreading the manuscript and clarification.
Author Response
Thank you very much for your comments
Round 3
Reviewer 2 Report
I have no substantive comments. The work needs intensive linguistic revision and the abstract should be rewritten.
My suggestion to change the title „Response of aerial and underground parts of different potato (Solanum tuberosum L.) cultivars to high temperature stress”
